# The Prevalence of Incidental Endometriosis in Women Undergoing Laparoscopic Ovarian Drilling for Clomiphene-Resistant Polycystic Ovary Syndrome: A Retrospective Cohort Study and Meta-Analysis

**DOI:** 10.3390/jcm8081210

**Published:** 2019-08-14

**Authors:** Marlene Hager, René Wenzl, Sonja Riesenhuber, Julian Marschalek, Lorenz Kuessel, Daniel Mayrhofer, Robin Ristl, Christine Kurz, Johannes Ott

**Affiliations:** 1Department of Obstetrics and Gynecology, Clinical Division of Gynecologic Endocrinology and Reproductive Medicine, Medical University of Vienna, Spitalgasse 23, 1090 Vienna, Austria; 2Department of Obstetrics and Gynecology, Clinical Division of Gynecologic Oncology and General Gynecology, Medical University of Vienna, Spitalgasse 23, 1090 Vienna, Austria; 3Section for Medical Statistics, Center for Medical Statistics, Informatics, and Intelligent Systems, Medical University of Vienna, Spitalgasse 23, 1090 Vienna, Austria

**Keywords:** polycystic ovary syndrome, endometriosis, ovarian drilling, laparoscopy, transvaginal hydrolaparoscopy

## Abstract

To evaluate the incidence of endometriosis in polycystic ovary syndrome (PCOS) patients who did not present with any endometriosis symptoms and underwent laparoscopic ovarian drilling (LOD) for clomiphene citrate (CC) resistance, 225 and 630 women with CC-resistant PCOS without classic endometriosis symptoms were included in a retrospective study and a meta-analysis, respectively. All women underwent LOD. The main outcome parameter was the prevalence of incidental endometriosis. Laparoscopy revealed endometriosis in 38/225 (16.9%) women (revised American Fertility Society (rAFS) stage I: 33/38, 86.8%; rAFS stage II: 5/38, 13.2%). When women with CC-resistant PCOS without endometriosis were compared, lower body mass index (BMI) and lower 25-hydroxy-vitamin D levels were associated with the presence of endometriosis at laparoscopy (odds ratios (OR): 0.872, 95% confidence intervals (95%CI): 0.792–0.960; *p* = 0.005 and OR: 0.980, 95%CI: 0.962–0.999; *p* = 0.036; respectively). The inclusion criteria for the meta-analysis were fulfilled by 4/230 reports about LOD. After correction for study heterogeneity, the pooled prevalence of incidental endometriosis was 7.7% in women with CC-resistant PCOS. In conclusion, the rate of incidental endometriosis in women with CC-resistant PCOS might reflect the prevalence of asymptomatic endometriosis. All cases were affected by minimal or mild disease. Since the literature lacks reports on associated clinical outcomes, the relevance of this entity in such patients should be the subject of further studies.

## 1. Introduction

Polycystic ovary syndrome (PCOS) and endometriosis are diseases that often occur among women of reproductive ages [1,2]. Both have a negative impact on fertility [2]. The treatment for PCOS in oligo- or anovulation in women who wish to have children is stimulation with either letrozole or clomiphene citrate (CC) [1]. A second- or third-line option in infertility treatment of PCOS is ovarian drilling, which is usually done via laparoscopy (laparoscopic ovarian drilling, LOD) [1]. Many studies have evaluated fertility outcome after LOD [3]. Given the fact that endometriosis seems to negatively affect oocyte quality [4], one might hypothesize that the success of LOD was influenced by the presence of concomitant endometriosis. Notably, only a few studies on LOD outcome have also included information about laparoscopically confirmed endometriosis [5,6,7]. 

In the clinical routine, infertile women with PCOS who report symptoms suspicious for endometriosis likely undergo laparoscopy, including ovarian drilling, early, probably even as a first-line treatment. This would also be in line with recent recommendations [1]. Accordingly, in women with PCOS who undergo LOD as a traditional second- or third-line treatment, the finding of endometriosis, if any, would be incidental. It seems worth mentioning that “asymptomatic endometriosis” has been defined “as endometriosis without pelvic pain and/or infertility” [2]. Strictly speaking, the lack of pain symptoms applies to these patients, but the lack of infertility cannot. However, since oligo- or anovulatory PCOS might be seen as the major contributing factor, the impact of endometriosis on fertility outcome in these patients remains unclear. Thus, it seems even more peculiar that the prevalence of endometriosis in women with PCOS has not been evaluated extensively. 

Moreover, during the last few years there has been increasing evidence that LOD can also be done efficiently via transvaginal hydrolaparoscopy. A huge advantage over laparoscopic access is that it can be performed in an outpatient setting and is associated with lower postoperative pain [8]. However, only the pouch of Douglas, the ovaries, and the tubes can be evaluated with this technique, whereas a big part of the intraperitoneal space cannot be visualized. Thus, one might miss endometriotic lesions or other pathologies, which would most likely not occur during laparoscopy [9,10,11]. Thus, the question arises whether transvaginal hydrolaparoscopy would be suitable for PCOS patients. Notably, complete resection of minimal to mild endometriosis is usually thought to improve the chance of natural conception [2,12], which is the main goal of LOD. Then, only the hydrolaparoscopic approach would be suitable, if no or only few cases of endometriosis would be missed. Of course, this would have to be evaluated in women who meet the criteria for transvaginal hydrolaparoscopy, namely, those with an anteverted-anteflected (avfl) uterus, who present without suspicion of major adhesions or endometriosis-associated symptoms [13]. 

Interestingly, it has been shown that anti-Mullerian hormone (AMH) has a destructive effect on the endometrium and on endometriosis cells [14,15]. It is well known that PCOS patients frequently reveal increased AMH-levels [16]. Moreover, as shown in the recent literature, DHEA-S levels—which were also elevated in many PCOS patients [16]—were also associated with endometriosis [17,18]. Last but not least, both diseases are linked with a lack of vitamin D and sometimes with hyperprolactinemia [19,20,21]. Thus, it would be of interest to know whether these serum marker levels would correlate with the prevalence of endometriosis and/or its severity in vivo and would, thus, potentially serve as predictors for the presence of endometriosis in women with asymptomatic PCOS.

Given all these considerations above and the lack of studies on this topic, the main objective of our study was to evaluate the prevalence of endometriosis in PCOS patients who did not present with any endometriosis symptoms and who underwent laparoscopic ovarian drilling for CC resistance. In addition, we also wanted to include data from our study and from previous reports into a meta-analysis to provide a sound overview. As secondary study aims, we also evaluated the predictive value of AMH, DHEAS, and 25-hydroxy vitamin D levels for the presence of endometriosis in this special patient population.

## 2. Material and Methods

### 2.1. Patient Population and Study Design of the Retrospective Cohort

In a retrospective cohort study, CC-resistant women with PCOS who underwent LOD at the Department of Gynecology and Obstetrics of the Medical University of Vienna, Vienna, Austria, between January 2008 and December 2018 were included (*n* = 240). PCOS was diagnosed according to the revised European Society of Human Reproduction and Embryology (ESHRE) and American Society for Reproductive Medicine (ASRM) criteria of 2004, which were based on the Rotterdam criteria [22,23]. All women revealed ≥12 follicles of 2–9 mm diameter on at least one ovary on transvaginal ultrasound, as well as 17-hydroxy progesterone levels <2 ng/mL, and, thus, non-classical adrenogenital syndrome could be excluded. LOD was performed by standard laparoscopy in all cases for clomiphene citrate resistance, which was defined as the absence of developing follicles after ovarian stimulation with 150 mg clomiphene citrate/day given for five days beginning with the fourth or fifth day of the menstrual cycle. Patients had been stimulated with clomiphene for a minimum of three and a maximum of six cycles.

In order to include women who were eligible for transvaginal hydrolaparoscopy, only patients who did not present classic endometriosis symptoms were selected. Thus, patients with moderate or severe dysmenorrhea (numeric rating scale ≥4) and women with dyspareunia were excluded (*n* = 6) [24]. Only a few patients showed sonographic suspicion of endometriomas (*n* = 8) or deep infiltrating endometriosis (*n* = 1) and were also excluded, since these women had been recommended to primarily undergo laparoscopy with concomitant LOD rather than first-line CC treatment, which is in line with current recommendations [1]. However, the abovementioned patients were also excluded, since we aimed to evaluate the prevalence of incidental endometriosis. All women had an avfl uterus and, thus, would have been eligible for transvaginal hydrolaparoscopy. This resulted in a patient population of 225 women.

The primary objective of the study was to evaluate the prevalence of incidental endometriosis. The latter was either confirmed laparoscopically and histologically or excluded laparoscopically. Secondarily, data on the site of endometriosis and the revised American Fertility Society (rAFS) score were collected [25], as well as whether the endometriosis lesions would have been accessible for the transvaginal hydrolaparoscopic approach, which was assumed if the lesion was located on the ovaries, the tubes, the posterior part of the uterus, and/or the peritoneum of the pouch of Douglas [26]. These outcome parameters were surgery specific and were retrieved by retrospective review of the surgical reports. Moreover, we focused on possible predictive parameters for the presence of endometriosis. In addition to the primary outcome parameters, we included the following data: patients’ age and body mass index (BMI) at the time of surgery; preoperative basal serum levels (one to two months before the operation) of luteinizing hormone (LH), follicle stimulating hormone (FSH), testosterone, DHEA-S, and AMH; 25 hydroxy-vitamin D before any supplementation with vitamin D (one to six months before the operation); and Fallopian tube patency.

The study was approved by the Institutional Review Board (IRB) of the Medical University of Vienna (IRB number 2088/2016). Data in this retrospective study were anonymized; thus, there was no need for informed consent according to the regulations of the IRB. There was no funding.

### 2.2. Laboratory Analyses in the Retrospective Cohort

Preoperative blood samples were taken from a peripheral vein between seven days and three months before LOD. All hormonal parameters were retrieved on the second to fifth cycle day. All examined serum parameters were determined in the central laboratory of the General Hospital of Vienna, Vienna, Austria using commercially available assays: Testosterone, ELECSYS^®^ Testosterone II, Roche Diagnostics GmbH, Mannheim, Germany; androstenedione, IMMULITE^®^ 2000 Androstenedione, Siemens Healthcare Diagnostics Products Ltd., Llanberis, UK; DHEA-S, ELECSYS^®^ DHEA-S, Roche Diagnostics GmbH, Mannheim, Germany; and AMH: DSL Active MIS/AMH assay; Beckman Coulter Inc., Brea, CA, USA.

### 2.3. Meta-Analysis

For the systematic literature review, we searched the Medline database (search date: 13 March 2019; search terms: Laparoscopic ovarian drilling) to identify cohort studies, systematic reviews, and meta-analyses about laparoscopic ovarian drilling, regardless of the drilling method used. Using the search term “laparoscopic ovarian drilling,” 230 articles were identified. The following reports were excluded step by step: Articles not published in the English language (*n* = 18); reviews without original data (*n* = 68); studies that included women with transvaginal hydrolaparoscopy only (*n* = 15); retracted articles and retraction notes (*n* = 4); articles that did not report the prevalence of endometriosis in their cases of laparoscopic ovarian drilling (*n* = 112); studies that did not provide data on endometriosis as an entity of its own, but summarized endometriosis cases with other factors for infertility (*n* = 3); and studies with endometriosis as an exclusion criterion (*n* = 4). Moreover, we excluded one study of our own working group whose patients were also included in the present analysis (*n* = 1) [5], and one study that focused on LOD as a tool to reduce the risk of re-occurrence of severe ovarian hyperstimulation syndrome (OHSS) and provided information about endometriosis only as an indication for in vitro-fertilization (IVF) [27]. Thus, four studies were included in the meta-analysis [6,7,28,29], in addition to the present case series. In all of these studies, PCOS had been diagnosed according to the revised Rotterdam criteria [22,23]. Three authors assessed the eligibility of the studies, extracted data on endometriosis prevalence, and assessed the risk of bias (M.H., D.M., and J.O.). Missing information and additional trials were not sought from authors.

### 2.4. Statistical Analysis

Variables are described by numbers (frequencies) and mean ± standard deviation. Statistical analysis was performed with SPSS 25.0 for Windows (SPSS Inc., 1989–2019) using the Fisher’s exact test for categorical parameters. Univariate binary logistic regression models were used to test the predictive value of all coefficients for the presence of endometriosis. Significant parameters were entered in a multivariate logistic regression model. Odds ratios (OR) and their 95% confidence intervals (95% CI) are given. Differences were considered statistically significant if *p* < 0.05.

For the meta-analysis on the prevalence of incidental endometriosis in women with PCOS undergoing laparoscopy, the library “metafor” in the open source statistical package “*R*” (The *R* Project for Statistical Computing) was used. The observed proportions were transformed using the Freeman–Tukey double arcsine transformation, which provides an effect measure with a favorable sampling distribution and stable variance. A meta-analysis model was fit to the transformed data using inverse variance weights and including a random effect to account for between-study heterogeneity. The random effects model was used due to the differences in observed endometriosis prevalence in the included studies, in order not to underestimate the variability of data. A pooled estimate of the prevalence of endometrioses and the corresponding 95% confidence interval were obtained by back-transforming the respective quantities to the original scale. Moreover, a leave-one-out sensitivity analysis and a funnel plot analysis to rule out publication bias were performed.

## 3. Results

### 3.1. The Prevalence of Incidental Endometriosis in Women with CC-Resistant PCOS: Results of the Retrospective Cohort

At the time of surgery, patients were 28.3 ± 5.0 years of age, with a BMI of 25.0 ± 4.8 kg/m^2^. Sterility was primary in 152 cases (67.6%). Laparoscopy revealed endometriosis in 38 women (16.9%). For the majority of these, a rAFS stage I was found (33/38, 86.8%), and in the remaining affected women the stage was II (5/38, 13.2%). Small peritoneal implants of the pelvic wall, on the urinary bladder, in the pouch of Douglas, or on the sacrouterine ligaments were found in 24 (63.2%), 13 (34.2%), 18 (47.4%), and 10 (26.3%) women, respectively. Of the 38 women with endometriosis, not all endometriosis lesions would have been accessible for the transvaginal approach in 22 (57.9%). 

### 3.2. The Prevalence of Incidental Endometriosis in Women with CC-Resistant PCOS: Results of the Meta-Analysis

Details of the meta-analysis are shown in Table 1. Two of the studies reported the presence of endometriosis since it was an exclusion criterion for further analysis [28,29]. The five studies included 630 women with PCOS, all of them CC resistant. After correction for study heterogeneity, the estimated endometriosis prevalence was 7.7% (Figure 1). Differences in study design, namely the retrospective and prospective approaches used (Table 1), might have introduced bias. Moreover, especially due to the fact that the majority of studies about LOD did not address endometriosis, publication bias must be assumed. The corresponding funnel plot is shown in Figure 2. All studies were plotted near the average. In the leave-one-out sensitivity analysis, the estimated endometriosis prevalence ranged from 5.9% to 10.1% (Table 2). In addition, the pooled prevalences without the random-effect model are shown in Table 2.

### 3.3. Additional Findings: Fallopian Tube Patency and Predictive Parameters for Endometriosis

Of 450 Fallopian tubes, 36 (8.0%) were found to be occluded during laparoscopy. Although bilateral tubal occlusion had been ruled out before CC stimulation in all patients by hysterosalpingography or hysterosalpingo-contrast sonography, which were performed 4–12 months before surgery, five women (2.2%) revealed bilateral occlusion at the time of LOD. This was significantly more common in women with endometriosis (3/38, 7.9%) than in those without it (2/185, 1.1%; *p* = 0.035).

When women with CC-resistant PCOS with and without endometriosis were compared to each other in univariate binary regression models (Table 3), lower BMI and lower 25-hydroxy-vitamin D levels were associated with the presence of endometriosis at laparoscopy. When these were entered into a multivariate model, they both remained statistically significant (OR: 0.872, 95%CI: 0.792–0.960; *p* = 0.005 and OR: 0.980, 95%CI: 0.962–0.999; *p* = 0.036; respectively).

## 4. Discussion

This analysis revealed three major findings. (i) There was a prevalence of incidental endometriosis in our population of CC-resistant PCOS patients of 16.9% and a prevalence of 7.7% in the meta-analysis. (ii) In our retrospective data set, all women with endometriosis had minimal or mild disease (rAFS stages I and II), with the vast majority having rAFS stage I (86.8%). (iii) Only a lower BMI and lower 25-hydroxy-vitamin D levels were predictive for endometriosis in our data. 

### 4.1. Incidental Endometriosis in Women with CC-Resistant PCOS and Its Possible Clinical Relevance

According to the meta-analysis, 7.7% of women with CC-resistant PCOS revealed endometriosis. If the pooled prevalence (Table 1) were directly calculated, which would be in accordance with a fixed effects model, a rate of 11.0% would be found. However, by considering study heterogeneity the estimated prevalence of incidental endometriosis in women with PCOS was 7.7%. This mentioned heterogeneity might hypothetically be due to differences in definition of CC resistance, laparoscopic diagnosis of endometriosis, and less familiar factors. The latter might include differences in patient characteristics. For example, other studies than ours included more women with a BMI > 30 kg/m^2^ [6,28,29]. This might have introduced minor deviances, as it could have been the case with differences in study quality. However, in the leave-one-out analysis (Table 2), the estimated endometriosis prevalence ranged from 5.9% to 10.1% which points toward only minor influences. Last but not least, despite the thorough literature search, studies published in languages other than English were excluded (*n* = 18) and, thus, some data might have been missed.

This prevalence of 7.7% could be compared to the prevalence of completely asymptomatic endometriosis found in women undergoing laparoscopic sterilization. While the latter ranged from 3% to 45% [30,31,32,33,34,35], our data seem quite comparable to those of the most recently published study (11%) [35]. Since PCOS and endometriosis do not share any pathophysiologic pathways, but are associated only with altered prolactin, DHEAS, and vitamin D levels [16,17,18,19,20,21], it seems reasonable that PCOS patients without endometriosis-suggestive symptoms would be burdened by endometriosis as often as other asymptomatic women. We consider this finding rather unsurprising and according to what one might expect. This obviously raises the question whether the finding of endometriosis would be of clinical relevance. Usually, asymptomatic endometriosis is defined as “endometriosis without pelvic pain and/or infertility” [2]. The latter does not apply to women with CC-resistant PCOS, but the literature lacks information about the impact of endometriosis on PCOS/LOD treatment outcome, and, thus, we consider this issue of importance for future studies.

Hand-in-hand with the discussion about the clinical relevance of endometriosis in this special patient population goes the severity of endometriosis. Only in our data set has this been evaluated in detail. It is noteworthy that the vast majority of women revealed minimal (rAFS I) and only about 13% of women had mild endometriosis (rAFS II). Thus, at least, we do not have to expect severely affected patients who would be in need of IVF treatment [2]. Thus, natural conception should be feasible in these women, which is also the main goal of LOD. Usually, in infertile women with mild to moderate endometriosis, it is recommended that clinicians perform operative laparoscopy with excision or ablation of the endometriosis lesions and adhesiolysis to improve fertility with the aim of natural conception [2]. More recent studies have even focused on the role of laparoscopy in enabling natural conception and avoiding IVF overuse. In 58% of women with recurrent implantation failure after IVF for unexplained infertility, incidental endometriosis was found. After complete resection, about 50% were able to conceive naturally within one year [36]. Moreover, in a randomized study of over 300 women with incidental minimal or mild endometriosis, operative laparoscopy led to fecundity rates of 4.7 per 100 person-months compared to 2.4 after diagnostic laparoscopy [37]. This suggests that complete resection of endometriosis might have a substantial role even in pain-asymptomatic women. In addition, another factor is the influence of endometriosis on early miscarriage, which is known to be greater in women with PCOS [38]. Miscarriage rates were reportedly higher in endometriosis patients than in disease-free women [39]. On the other hand, a randomized study suggested that complete resection of minimal or mild endometriosis was associated with one-year birth rates similar to only diagnostic laparoscopy [40]. Thus, knowledge about the presence of incidental endometriosis in women with CC-resistant PCOS might not alter the treatment strategy at this time, although removal of the lesions might be endorsed by some experts. In other words, it seems up to the individual surgeon how to deal with this situation at present. 

### 4.2. Endometriosis Prevalence in Women Undergoing LOD and Its Relevance for the Transvaginal Hydrolaparoscopic Approach

The considerations above lead to the question of whether it would be reasonable to perform transvaginal hydrolaparoscopy for LOD. Due to the fact that only few lesions would be expected and that the majority of these would be located outside the area accessible for a transvaginal approach, many lesions would be missed. Notably, there is only indirect evidence. In terms of postoperative AMH reduction, the techniques of standard LOD and drilling via transvaginal hydrolaparoscopy seem comparable [11]. However, there are no randomized reports about clinical outcome, namely, pregnancy and live-birth rates. One study compared another drilling method that could not assess flat endometriotic lesions, i.e., ultrasound-guided transvaginal ovarian needle drilling, to standard LOD: Patients in the LOD group experienced a significantly lower AMH, lower antral follicular count, higher ovulation rate, and higher pregnancy rate after six months [41]. It seems reasonable that the lower chance of ovulation contributed widely to the lower pregnancy rate in the transvaginal ovarian needle-drilling group. However, the possible consequence of missed endometriotic lesions cannot be completely denied, especially since endometriosis had been an exclusion criterion in this study, which applied only to sonographically detectable endometriosis and to those women who underwent LOD, to the best of our understanding. Last but not least, it seems noteworthy that the non-inferiority of transvaginal hydrolaparoscopic ovarian drilling has not yet been proven.

### 4.3. Fallopian Tube Patency and Predictive Parameters for Endometriosis

One other interesting finding was that, despite having ruled out bilateral Fallopian tube occlusion for all women in the course of routine diagnostic evaluation before CC stimulation, laparoscopy revealed this abnormality in five patients, with a significantly increased risk for women with endometriosis (7.9% versus 1.1%, respectively). Although bilateral tubal spasm cannot be ruled out completely [42], this would point to a clinical relevance of incidental endometriosis in PCOS women.

Both in the uni- and the multivariate model that evaluated parameters associated with incidental endometriosis in our women with CC-resistant PCOS, lower BMI and lower 25-hydroxy-vitamin D levels were significantly predictive. The odds ratios (0.872 and 0.980, respectively) suggest only minor clinical relevance. It has already been suggested that a higher BMI was associated with a lower endometriosis risk and that the underlying biologic mechanisms will have to be elucidated in the future [43]. A vitamin D deficiency, with its effects as an immunomodulator and anti-inflammatory agent, has been mentioned as a possible factor in endometriosis pathogenesis, but this relationship remains still unclear [44]. Notably, AMH levels were not associated with the presence of incidental endometriosis despite the suggested destructive effect of AMH on endometriosis cells [14,15]. All in all, none of the tested parameters provided clinically relevant information about the occurrence of endometriosis. However, due to the special patient population of the present study, characterized namely by CC-resistant PCOS and only incidental endometriosis, the data cannot be generalized.

### 4.4. Study Limitations

First, the retrospective design and the sample size of the retrospective case series need to be mentioned. The latter also led to the necessity of performing univariate analyses of risk factors for endometriosis presence first and including only significant parameters into the multivariate model. It might be argued that this approach, although widely used, was less reliable than including all parameters in a multivariate model at once. However, we consider the number of patients included in the meta-analysis sufficient despite the fact that the majority of studies on LOD have not mentioned endometriosis. Although all PCOS patients in the meta-analysis had suffered from CC resistance, the slight differences in PCOS and/or patient characteristics as well as in study quality have to be emphasized as discussed above. Moreover, diagnostic laparoscopy needs to be well-performed in order to accurately exclude endometriosis [2], which likely depends on the surgeon’s expertise and thoroughness. These circumstances might have contributed to the substantial variances in endometriosis rates found in the studies, for example the higher prevalence in our study population (Table 2). Furthermore, risk of publication bias must also be taken into account. Last but not least, neither the CC-stimulation regimen nor CC resistance was defined in detail in the majority of included studies [6,7,28,29]. In our data set, CC was started on cycle day 4 or 5, which is not in accordance with the majority of recently published studies [45]. Although this might be considered late, the results of CC stimulation are comparable when it was started on cycle day 2, 3, 4, or 5 [46,47].

### 4.5. Conclusions

Incidental endometriosis was found in 7.7% of women with CC-resistant PCOS according to the meta-analysis. Since this might reflect the prevalence of asymptomatic endometriosis and the literature lacks reports on associated clinical outcomes, its relevance remains open and should be the subject of further studies.

## Figures and Tables

**Figure 1 jcm-08-01210-f001:**
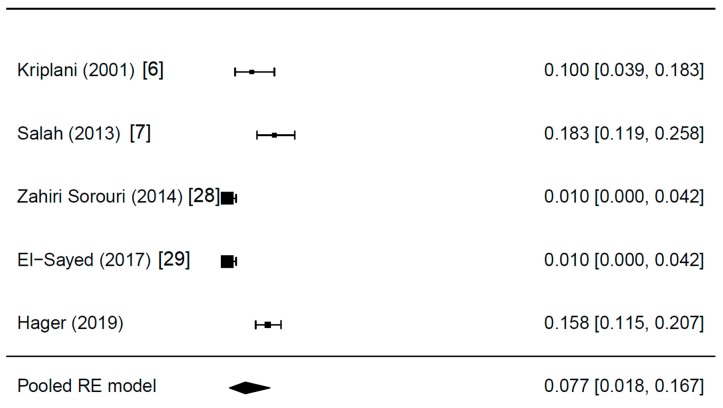
The prevalence of incidental endometriosis in women with CC-resistant PCOS: Results of the meta-analysis.

**Figure 2 jcm-08-01210-f002:**
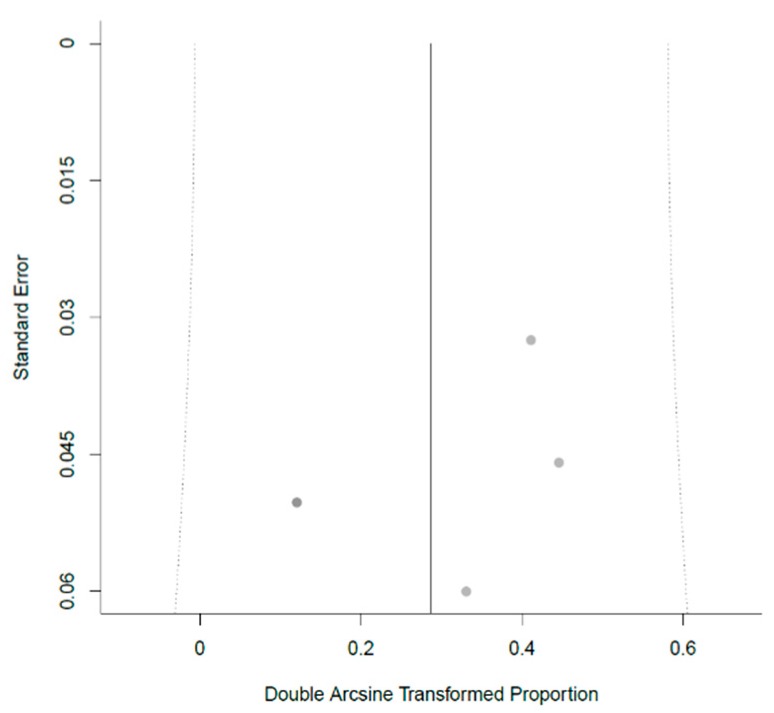
Funnel plot for the meta-analysis.

**Table 1 jcm-08-01210-t001:** Studies about endometriosis prevalence in women undergoing laparoscopic ovarian drilling included into the pooled analysis.

First Author (year)	Women with PCOS (*n*)	Study Design	Years	Characteristics of PCOS Patients	Women with Endometriosis (*n*, %)	Additional Findings
Kriplani (2001) [6]	70	Retrospective	?	Anovulatory, CC resistant	7 (10.0)	Endometriotic cyst in *n* = 4 (5.7%)
Salah (2013) [7]	120	Retrospective	?	Anovulatory, CC resistant	22 (18.3)	–
Zahiri Sorouri (2014) [28]	100	Prospective	2011–2012	CC resistant	1 (1.0)	–
El-Sayed (2017) [29]	100	Prospective	2015–2017	CC resistant, BMI < 30 kg/m^2^, LH >10 IU/mL or LH/FSH ratio >2, free androgen index >4, normal oGTT	1 (1.0)	–
Hager (2019)	240	Retrospective	2008–2018	Anovulatory, CC resistant, women without signs of endometriosis	38 (16.9)	–

Abbreviations used: CC, clomiphene citrate; BMI, body mass index; LH, luteinizing hormone; FSH, follicle stimulating hormone; oGTT, oral glucose tolerance test.

**Table 2 jcm-08-01210-t002:** Estimates and 95% confidence intervals obtained in a leave-one-out sensitivity analysis. The calculations are performed as in the main analysis. However, in each round of the sensitivity analysis one study is not included.

	Estimate Random Effect (95% CI)	Estimate Pooled (95% CI)
All studies included	0.077 (0.018; 0.167)	0.110 (0.085; 0.134)
Without Kriplani (2001) [6]	0.071 (0.007; 0.188)	0.111 (0.085; 0.137)
Without Salah (2013) [7]	0.056 (0.006; 0.145)	0.092 (0.067; 0.117)
Without Zahiri Sorouri (2014) [28]	0.101 (0.030; 0.204)	0.128 (0.100; 0.157)
Without El-Sayed (2017) [29]	0.101 (0.030; 0.204)	0.128 (0.100; 0.157)
Without Hager (2019)	0.059 (0.004; 0.159)	0.079 (0.053; 0.106)

**Table 3 jcm-08-01210-t003:** General patient characteristics and hormonal profile in women with CC-resistant PCOS with and without endometriosis.

	Women with Endometriosis (*n* = 38)	Women without Endometriosis (*n* = 187)	Univariate Analysis		Multivariate Analysis	
			Adjusted OR (95% CI)	*p*	Adjusted OR (95% CI)	*p*
Age (years) *	28.5 ± 4.8	28.3 ± 5.0	1.009 (0.941; 1.083)	0.792	–	–
Body mass index (kg/m^2^) *	23.3 ± 5.2	25.3 ± 4.6	0.905 (0.830; 0.987)	0.024	0.872 (0.792; 0.960)	0.005
Primary sterility ^#^	30 (78.9)	122 (65.2)	0.501 (0.217; 1.155)	0.105	–	–
Thyroid stimulating hormone (IU/mL) *	1.6 ± 1.0	1.7 ± 1.0	0.888 (0.587; 1.343)	0.573	–	–
Prolactin (ng/mL)	11.7 ± 5.1	13.5 ± 7.1	0.954 (0.896; 1.017)	0.151		
25 OH vitamin D (nmol/L)	35.0 ± 24.8	43.8 ± 22.3	0.982 (0.965; 0.999)	0.049	0.980 (0.962; 0.999)	0.036
LH (mIU/mL) *	11.6 ± 7.5	12.6 ± 9.9	0.987 (0.945; 1.031)	0.553	–	–
FSH (mIU/mL) *	5.7 ± 2.3	5.6 ± 2.0	1.019 (0.851; 1.219)	0.838	–	–
Testosterone (ng/mL) *	0.43 ± 0.24	0.48 ± 0.25	0.432 (0.080; 2.344)	0.331	–	–
DHEAS (µg/mL)	2.60 ± 1.20	2.57 ± 1.15	1.020 (0.732; 1.420)	0.909	–	–
AMH (ng/mL) *	9.1 ± 6.2	9.5 ± 7.6	0.992 (0.935; 1.052)	0.783	–	–

Data are presented as * mean ± standard deviation for numerical parameters or as ^#^ numbers (frequencies) for categorical parameters. Abbreviations used: OR, odds ratio; AMH, anti-Mullerian hormone.

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
