# Peer review of "The Prevalence of Incidental Endometriosis in Women Undergoing Laparoscopic Ovarian Drilling for Clomiphene-Resistant Polycystic Ovary Syndrome: A Retrospective Cohort Study and Meta-Analysis"

_jcm, 2019, doi:10.3390/jcm8081210_

Round 1

Reviewer 1 Report

This report on the incidence of endometriosis in clomid resistant PCOS women experiencing laparoscopic drilling for their infertility, describes an incidence between 8 and 16%, depending on the two types of studies analyzed, and most exhibited mild disease. The authors, however, do not seem too sure of what to make of these findings, apart from suggesting more research is needed. Is this incidence greater than expected? If not, what is the clinical significance of knowing? Would clinical practice change on knowing that such mild, asymptomatic endometriosis was present? Inclusion of such discussion is encouraged. About a 50% reduction of detailed discussion of prior studies is also encouraged.

Table 3. Testosterone levels should be in ng/ml not pg/ml.

Author Response

Dear Reviewer,

We thank you for your comments and recommendations regarding our manuscript “The prevalence of incidental endometriosis in women undergoing laparoscopic ovarian drilling for clomiphene-resistant polycystic ovary syndrome: A retrospective cohort study and meta-analysis”. We took care in revising our work according to your suggestions and hope that the revisions made will make our manuscript acceptable for publication in the Journal of Clinical Medicine.

Please find the Point-by-Point answer letter below. The manuscript was proof-read by an English native speaker.

We thank you for your interest in our manuscript,

Respectfully yours,

Johannes Ott

Reviewer's comments:

This report on the incidence of endometriosis in clomid resistant PCOS women experiencing laparoscopic drilling for their infertility, describes an incidence between 8 and 16%, depending on the two types of studies analyzed, and most exhibited mild disease. The authors, however, do not seem too sure of what to make of these findings, apart from suggesting more research is needed. Is this incidence greater than expected? If not, what is the clinical significance of knowing? Would clinical practice change on knowing that such mild, asymptomatic endometriosis was present? Inclusion of such discussion is encouraged.

Reply:

Thank you for your comment. We included the following considerations to the Discussion Section: “We consider this finding rather unsurprising and according to what one might expect.”

“Thus, knowledge about the presence of incidental endometriosis in CC-resistant PCOS women might not alter the treatment strategy at this time, although removal of the lesions might be endorsed by some experts. In other words, it seems up to the individual surgeon how to deal with this situation at present.”

About a 50% reduction of detailed discussion of prior studies is also encouraged.

Reply:

We tried to shorten the paragraph about “Incidental endometriosis in CC-resistant PCOS women and its possible clinical relevance” (4.1). We were able to delete about 150 words which is a reduction of over 20% of this paragraph. Moreover, one reference was deleted. Honestly, we believe that the detailed information provided in the Discussion Section helps the reader to put the findings into a clinically relevant context. Moreover, Reviewer number 2 strongly recommended to extend the Discussion Section. We hope that the revisions made will make the manuscript acceptable for you. In case of any further queries, we shall be happy to revise our work a second time.

Table 3. Testosterone levels should be in ng/ml not pg/ml.

Reply:

Sorry for the typo. Corrected.

Reviewer 2 Report

This article reports on the incidence of mild and minimal endometriosis when drilling for PCOS

Drilling is considered in cas of resistance after cycles of clomiphène, but in this cases, clomiphene is started quite late (4-5th day) line 102)

Reading this article, I still understand if drilling was done by laparoscopy or trans-vaginal hydrolaparoscopy ? If it's not laparoscopy, the chapter (lines 62-74) has no reason to be written

On the other hand, the chapters linking PCOS and endometriosis (AMH, DHEA), as well prolactin are interesting

Let's stop clicking only Marcoux study, whose study is biased. the italian's study showing the lack of benefit of surgical treatment of mild and minimal endometriosis (ref 39, line 279)

Article to be published, minor revisions

Author Response

Dear Reviewer,

We thank you for your comments and recommendations regarding our manuscript “The prevalence of incidental endometriosis in women undergoing laparoscopic ovarian drilling for clomiphene-resistant polycystic ovary syndrome: A retrospective cohort study and meta-analysis”. We took care in revising our work according to your suggestions and hope that the revisions made will make our manuscript acceptable for publication in the Journal of Clinical Medicine.

Please find the Point-by-Point answer letter below. The manuscript was proof-read by an English native speaker.

We thank you for your interest in our manuscript,

Respectfully yours,

Johannes Ott

Reviewer's comments:

This article reports on the incidence of mild and minimal endometriosis when drilling for PCOS

Drilling is considered in case of resistance after cycles of clomiphène, but in this cases, clomiphene is started quite late (4-5th day) line 102)

Reply:

We agree, that according to a recent review (Gadalla et al. 2018) many studies started on cycle day 3. However it has been claimed that the results are comparable when CC is started on cycle day 2, 3, 4 or 5 (Wu and Winkel 1989, Dehbashi et al. 2006). In terms of ovulation, pregnancy and miscarriage rates. Nonetheless, we added the following statement to the Discussion Section (study limitations): "Last but not least, neither CC stimulation regimen nor CC-resistance was defined in detail in the majority of included studies [6, 7, 29, 30]. In our data set CC was started on cycle day 4 or 5, which is not in accordance with the majority of recently published studies. [47] Although this might be considered late, the results of CC stimulation are comparable when it is started on cycle day 2,3,4 or 5. [48,49]"

We added the following references:

Gadalla MA, Huang S, Wang R, Norman RJ, Abdullah SA, El Saman AM et al. Effect of clomiphene citrate on endometrial thickness, ovulation, pregnancy and live birth in anovulatory women: systematic review and meta-analysis. Ultrasound Obstet Gynecol 2018;51(1):64-76. Wu CH, Winkel CA. The effect of therapy initiation day on clomiphene citrate therapy. Fertil Steril 1989;52:564. Dehbashi S, Vafaei H, Parsanezhad MD, Alborzi S. Time of initiation of clomiphene citrate and pregnancy rate in polycystic ovarian syndrome. Int J Gynaecol Obstet 2006;93:44.

Reading this article, I still understand if drilling was done by laparoscopy or trans-vaginal hydrolaparoscopy ? If it's not laparoscopy, the chapter (lines 62-74) has no reason to be written

Reply:

Thank you for the comment. We clarified this issue in the Methods Section as follows: "LOD was performed by standard laparoscopy in all cases for clomiphene citrate resistance, which was defined as the absence of developing follicles after ovarian stimulation with 150mg clomiphene citrate/day given for five days beginning with the 4th or 5th day of the menstrual cycle."

On the other hand, the chapters linking PCOS and endometriosis (AMH, DHEA), as well prolactin are interesting

Reply:

We thank you for your appreciation.

Let's stop clicking only Marcoux study, whose study is biased. The Italian’s study showing the lack of benefit of surgical treatment of mild and minimal endometriosis (ref 39, line 279)

Reply:

Thank you for your input. We added the following statement to the Discussion Section, that should lessen the importance of the Marcoux study: “On the other hand, a randomized study suggested that complete resection of minimal or mild endometriosis was associated with one-year birth rates similar to only diagnostic laparoscopy” [42].

Reviewer 3 Report

Some of the number of the tables described in the text do not coincide with the reciprocal number of the Tables. They are as follows: 

           line 194, Table 2 should be Table 1 

           line 198, Table 3 should be Table 2

           line 220, Table 1 should be Table 3

           line 236, Table 3 should be Table 1

           line 245, Table 3 should be Table 2

The prevalence of incidental endometriosis in this study should be 16.9% (38/225) rather than 15.8%  In section 4.3 of Discussion, the conclusion of ref. 44 is actually higher (not lower) body mass index may be associated with lower risk of endometriosis. The authors made an opposite description. So the result of this study is actually compatible with that of ref.44 (Table 3).

        In addition, the authors quoted the ref 45 to conclude that Vit. D deficiency is a risk factor for endometriosis. However, that paper was published in 2014. According to recent studies, the relationship between Vit. D deficiency and endometriosis is still not clear. The  authors are suggested to consult the following reports: 

Is there a relationship between Vitamin D and endometriosis? An overview of literature. (Curr Pharm Des 2019 Jul 21) 25-hydroxyvitamin D serum levels and endometriosis: results of a case-control study.  (Reprod Sci 2019 Feb;26(2):172-7)  

        Therefore, all the parameters in Table 3 seem to provide no information about the occurrence of endometriosis in CC-resistant PCOS patients. Because lower BMI is an already known risk factor for endometriosis no matter if PCOS is present.  

Author Response

Dear Reviewer,

We thank you for your comments and recommendations regarding our manuscript “The prevalence of incidental endometriosis in women undergoing laparoscopic ovarian drilling for clomiphene-resistant polycystic ovary syndrome: A retrospective cohort study and meta-analysis”. We took care in revising our work according to your suggestions and hope that the revisions made will make our manuscript acceptable for publication in the Journal of Clinical Medicine.

Please find the Point-by-Point answer letter below. The manuscript was proof-read by an English native speaker.

We thank you for your interest in our manuscript,

Respectfully yours,

Johannes Ott

Reviewer 3:

Some of the number of the tables described in the text do not coincide with the reciprocal number of the Tables. They are as follows:

           line 194, Table 2 should be Table 1

           line 198, Table 3 should be Table 2

           line 220, Table 1 should be Table 3

           line 236, Table 3 should be Table 1

           line 245, Table 3 should be Table 2

Reply: We are sorry for these mistakes. Corrected as recommended.

The prevalence of incidental endometriosis in this study should be 16.9% (38/225) rather than 15.8% In section 4.3 of Discussion, the conclusion of ref. 44 is actually higher (not lower) body mass index may be associated with lower risk of endometriosis. The authors made an opposite description. So the result of this study is actually compatible with that of ref.44 (Table 3).

Reply: Again, we are sorry for the unnecessary mistakes.

- 18.5% was corrected to 16.9% throughout the manuscript.

- The wording in the mentioned Section of the Discussion was changed as follows: “It has already been suggested that a lower higher BMI was associated with a lower endometriosis risk and that the underlying biologic mechanisms will have to be elucidated in the future [44].” You are correct, now the results are comparable.

In addition, the authors quoted the ref 45 to conclude that Vit. D deficiency is a risk factor for endometriosis. However, that paper was published in 2014. According to recent studies, the relationship between Vit. D deficiency and endometriosis is still not clear. The authors are suggested to consult the following reports:

Is there a relationship between Vitamin D and endometriosis? An overview of literature. (Curr Pharm Des 2019 Jul 21) 25-hydroxyvitamin D serum levels and endometriosis: results of a case-control study.  (Reprod Sci 2019 Feb;26(2):172-7) 

Reply: We thank the reviewer for making us aware of this interesting refrence. We replaced the original reference number 45 with the new one and revised the according sentence in the Discussion Section as follows: “A vitamin D deficiency, with its effects as an immunomodulator and anti-inflammatory agent, has been mentioned as a possible factor in endometriosis pathogenesis, but this relationship remains still unclear [45].”

Therefore, all the parameters in Table 3 seem to provide no information about the occurrence of endometriosis in CC-resistant PCOS patients. Because lower BMI is an already known risk factor for endometriosis no matter if PCOS is present. 

Reply: Thank you for this comment. We took the liberty of adding your sentence to the Discussion Section (4.3): “All in all, none of the tested parameters provided clinically relevant information about the occurrence of endometriosis.”

In addition to making the minor revisions, I think the authors need to make some change in the discussion part.

Reply: We revised the Discussion Section, which was also in accordance with the other reviewers. In detail, we made the following changes:

- We added a comment to the direct clinical significance of our findings to Section 4.1: “We included the following considerations to the Discussion Section: “We consider this finding rather unsurprising and according to what one might expect.”

“Thus, knowledge about the presence of incidental endometriosis in CC-resistant PCOS women might not alter the treatment strategy at this time, although removal of the lesions might be endorsed by some experts. In other words, it seems up to the individual surgeon how to deal with this situation at present.”

- We shortened the paragraph about “Incidental endometriosis in CC-resistant PCOS women and its possible clinical relevance” (4.1). We were able to delete about 150 words which is a reduction of over 20% of this paragraph. Moreover, one reference was deleted.

- Concerning the starting day of CC stimulation, we added the following statement to the Discussion Section (study limitations): “Last but not least, neither CC stimulation regimen nor CC-resistance was defined in detail in the majority of included studies [6, 7, 29, 30]. In our data set CC was started on cycle day 4 or 5, which is not in accordance with the majority of recently published studies. [47] Although this might be considered late, the results of CC stimulation are comparable when it is started on cycle day 2,3,4 or 5. [48,49]”

- Last not least, we added a comment about the Italian study on the relevance of resection of minimal to mild endometriosis: “On the other hand, a randomized study suggested that complete resection of minimal or mild endometriosis was associated with one-year birth rates similar to only diagnostic laparoscopy [42].”

Round 2

Reviewer 3 Report

I think the revised manuscript is qualified for publication.